Trophic degradation predispositions and intensity in a high-flow, silted reservoir

Bartoszek Lilianna bartom@prz.edu.pl
Miąsik Małgorzata
Koszelnik Piotr
Department of Environmental Engineering and Chemistry, Faculty of Civil and Environmental Engineering and Architecture, Rzeszów University of Technology , Rzeszów , Poland
Sánchez-Carrillo Salvador
Electronic publication date: 2020 Jun 22
Publication date: 2020
Volume: 8
Electronic Location ID: e9374
Received 2020 Jan 13; Accepted 2020 May 27
Copyright: ©2020 Bartoszek et al.
Copyright year: 2020
Copyright holder: Bartoszek et al.
License: This is an open access article distributed under the terms of the Creative Commons Attribution License, which permits unrestricted use, distribution, reproduction and adaptation in any medium and for any purpose provided that it is properly attributed. For attribution, the original author(s), title, publication source (PeerJ) and either DOI or URL of the article must be cited.
License URL: https://creativecommons.org/licenses/by/4.0/

Keywords: Dam reservoir, Trophic degradation, EPC-0, Resilience to degradation, Impact of the catchment, Natural susceptibility to degradation

Funding: Polish National Science Center 2011/03/B/ST10/04998 The research was funded by the Polish National Science Center in the framework of the research project No. 2011/03/B/ST10/04998. Piotr Koszelnik received a research grant from the Polish National Science Center: project No. 2011/03/B/ST10/04998.

==============================
The objective of the work was to demonstrate the relationship between the natural environmental characteristics of a reservoir and its catchment and severity of trophic degradation. The shallow, highly-silted Rzeszów Reservoir (SE Poland) was the object of study. The impact on degradation of internal supply from accumulated bottom sediments was also assessed, using water and sediment sampled in 2013 and 2014. A high value for trophic state was identified for the reservoir on the basis of TSI indexes, while assessed natural resilience to degradation and analysis of the catchment as a supplier of biogenic and organic matter both indicate high susceptibility to cultural eutrophication. Obtained values for equilibrium phosphate concentrations under anoxic conditions (EPC-0) point to the possibility of a more intensive process of internal supply in phosphorus. However, the presence of sediments poor in organic matter suggest no major threat of ongoing eutrophication. Desludging and/or dredging are likely to entail elimination from the ecosystem of a large part of the pollutants accumulated in sediments, as well as the internal supply of phosphate to the water column. However, as external sources are responsible for the advanced degradation of Rzeszów Reservoir, any attempts at reclamation within the water will fail to yield persistent effects if appropriate protective procedures in the catchment are not implemented.

Introduction

The primary function of small reservoirs is to retain water and balance flow in the river below. The main economic problem characteristic for waters of this size is their tendency to experience ongoing deterioration in water quality, as associated with a rapid reduction in capacity and therefore declining resilience to supplied contaminants. A marked process of silting and shallowing is also associated with intensified sediment resuspension that combines with turbidity to encourage secondary pollution with substances accumulated in bottom sediments (Cyr, McCabe & Nürnberg, 2009; Tammeorg et al., 2016; Lee et al., 2019).

These problems reflect both sedimentation of suspended inflowing material and allochthonous river-sediment transport and trail along the reservoir bottom, as well as the production of matter within the reservoir via eutrophication (Michalec & Tarnawski, 2008; Dunalska, 2011; Wiatkowski & Rosik-Dulewska, 2016). Such factors responsible for degradation of the ecosystem often interact with increased strength due to a reservoir being located improperly. At the design stage the impact of a catchment is often overlooked, and most especially the relationship between the area of the latter and the surface area of the reservoir to be constructed. In many cases, there is a further failure to fully analyze environmental characteristics in the watershed, above all geology and topography, as well as the impact of agricultural land use (Grochowska, Brzozowska & Lopata, 2013; Koszelnik & Gruca-Rokosz, 2013).

Given their more favorable parameters as regards morphometry (including average depth and volume) and sometimes also hydrology (e.g., hydraulic retention time), larger bodies of water prove more resilient to eutrophication than small ones (Bartoszek & Czech, 2014; Bartoszek, 2019). Small and shallow reservoirs have what might be regarded as impaired defense mechanisms, hence the tendency for them to undergo rapid ongoing degradation. Furthermore, reservoirs located in heavily-urbanized areas are degraded much faster than those in forest, or in regions supporting agro-forestry (Gulati & Donk, 2002; Jachniak, 2011). A significant reason for this is the supply of major loads of pollutants from both point and non-point sources (Doig et al., 2017). The occurrence of internal phosphorus supply is also of great importance for the state of a reservoir, given the way trophic degradation is intensified, while the effectiveness of introduced protective activities in the catchment is limited (North et al., 2015; Tammeorg et al., 2016; Paytan et al., 2017). The internal load can equate to as much as 45 to 89% of the external load (Nürnberg et al., 2013). Phosphorus release from sediments occurs in a bioavailable form in reservoir water (as P-PO43−), with this ensuring its greater availability to phytoplankton than the P from external loading introduced mainly in the form of suspended solids (Loh et al., 2013; North et al., 2015; Doig et al., 2017). Most reclamation techniques applied currently focus on eliminating the internal supply of nutrients (mainly phosphates, only rarely nitrates), or else on limiting their actual impact on the aquatic ecosystem (Gulati & Donk, 2002; Hickey & Gibbs, 2009; Wojciechowska, Gajewska & Ostojski, 2017).

The parameter used to assess the capacity of bottom sediment to release P-PO43− is the value of the zero equilibrium phosphate concentration (EPC-0) (Wisniewski, 1999; Bartoszek & Koszelnik, 2016). Release of phosphate from sediment into the water column is to be expected where the near-bottom concentration of P-PO43− falls below the EPC-0. In turn, where the value exceeds it, adsorption on to particles of sediment is to be anticipated (House, 2003; Cyr, McCabe & Nürnberg, 2009; Dong, Yang & Liu, 2011). The choice of the right method of restoration is in turn dependent on characteristics of the given body of water, be these morphometric, or related to location within the catchment and actual pollutant loads (Malecki, 2005). Differences between reservoirs (notably involving hydrological conditions and depth) ensure that not all restoration methods can be applied to each object.

The work detailed here sought to determine the relationship between natural environmental characteristics of a reservoir and its catchment area and the severity of trophic degradation. The example for study with this objective was provided by the shallow, highly-silted Rzeszów Reservoir (SE Poland), whose internal supply from accumulated bottom sediments was also assessed for its degrading impact. In general, the reservoir is able to serve as a good example of bad practice where continuous conservation and reclamation are concerned. This reflects an imperfect location and consequent ongoing degradation.

While there are no reliable data, it is estimated that there may be in excess of 16 million small bodies of water around the world (Mulligan, Soesbergen & Sáenz, 2020). This fact has combined with often-advanced processes of degradation, and serious problems with remediation, to ensure a continuous history of research in this area.

Materials & Methods

Study area and sampling strategy

The researched Rzeszów Reservoir was built in 1974 through the damming of the River Wislok at a point 63+760 km along its course. The reservoir is supplied by two main tributaries, i.e., the Wislok and the Strug. Its main purpose was to allow for proper operations of the water supply to the city of Rzeszów. However, given a location on the outskirts of such a large city, a vital role as a sports and recreation lagoon is also served (Bartoszek et al., 2015). Morphometric parameters of the reservoir in 2014 are as shown in Fig. 1. Overall volume is seen to have decreased by about 40% over the 40-year period (from 1.8 ⋅106 to 1.1 ⋅106 m3), with major silting having taken place, and indeed a gradual development of new land surface in the upper zone in particular. Attempts to restore greater usability to the reservoir were made in the years 1986–87 and 1995–97, entailing work to deepen the reservoir next to the dam, while also achieving a narrowing through partial backfill on the right part of the bank also just by the dam. In each case, some 250,000–300,000 m3 of sediment were removed (Bartoszek et al., 2015). The objective of both dredging operations was to secure an increase in rate of flow, thereby reducing sedimentation. Unfortunately, this restoration did not bring the expected results. After just 7 years, sedimentation had exceeded the amount removed previously (Madeyski, Michalec & Tarnawski, 2008).

Figure 1 Localization of the Rzeszów Reservoir with its parameters and sampling stations.

As the watershed of Rzeszów Reservoir is of 2,061 km2, this represents a considerable proportion of the entire Podkarpackie voivodship (province-region of Poland). The difference in water level between the main supply of the Wislok at the source and the mouth of the reservoir is 616 m. The Wislok flows through foothill areas that are largely agricultural, though the upper parts are forested, while the middle part does also have industrial centers (glassworks, tanneries and refineries) along its course. The catchment of the smaller tributary, the Strug stream, is predominantly agricultural, with traditional fragmented patches of farmland typically associated with a high-density population. The share of individual land-use forms in the catchment is arable land −55%, built-up areas −10%, forest areas −20%, wasteland and meadows −15%. The reservoir can thus be said to be under strong anthropopressure associated with local agriculture that causes severe erosion of the land. Wastes of various kinds are also deposited in the area, and other kinds of diffuse pollution occur (Koszelnik, 2007; Gruca-Rokosz, Tomaszek & Koszelnik, 2009; Report, 2016).

Water and sediment samples were taken from three sites along the axis of the reservoir (Fig. 1), four times in the May-September period of 2013, and five times in the same period of 2014.

Rzeszów Reservoir’s natural susceptibility to degradation was evaluated on the basis of two procedures (described in more detail in File S1):

1. Resilience in the face of degradation was assessed by modifying the so-called Lake Quality Assessment Scheme (Bajkiewicz-Grabowska, 1987; Bajkiewicz-Grabowska, 2010) in line with the status of the reservoir as an artificial body of water.

2. Assessment of catchment-area impacts relating to the supply of the reservoir in biogenic and organic matter was carried out in line with the system proposed by Bajkiewicz-Grabowska (1987) and Bajkiewicz-Grabowska(2010), as well as Markowski & Kwidzinska (2015).

Water analysis

Temperature (Tw), pH and dissolved oxygen (O2) were measured in situ with a Hach Lange HQ40D meter. Total nitrogen (TN) was determined using a TOC-VCPN analyzer (Shimadzu), phosphate-phosphorus (P-PO43−) and chlorophyll a spectrophotometrically (Aquamate, Thermo Spectronic) using filtered samples of water following reaction with ammonium molybdate and hot extraction with ethanol, respectively. Total phosphorus (TP) was determined analogously, but in non-filtered and mineralized (H2SO4 and peroxodisulfate) samples of water. Trophic status of the water was approximated by reference to Carlson Indexes (TSITP and TSIChla) (Carlson, 1977).

Sediment analysis

Sediment samples (from the 0–5 cm layer) were dried and further assessed for Loss-on-Ignition (LOI) at 550 °C for four hours (4 h), with this construed as organic matter content (OM). For the analysis of P fractionation in sediment, the SMT (Standards, Measurements and Testing) method was customized (Pardo, Rauret & Lopez-Sanchez, 2004). The fractions of P obtained were non-apatite, inorganic (NAIP, associated with oxides and hydroxides of Al, Fe and Mn), apatite (AP, associated with Ca) and organic (OP). Following microwave mineralization (HNO3, 2–4.5 MPa), TP was measured as above, with quantitative determinations for Fe, Ca, Al and Mn made using an ICP spectrometer (Integra, GBC).

Measuring of the EPC-0

EPC-0 values were measured in aerobic and anoxic conditions in association with P-PO43− concentrations in above-surface water in the range 0.0 to 1.3 mgP dm−3 (as KH2PO4). For this purpose, six intact sediment cores were collected into Plexiglas tubes (from the thin-upper layer of 0–5 cm). Natural water was decanted and replaced by reservoir water diluted 1:10 with distilled water, this then being deoxygenated using anhydrous sodium sulfate (IV), in order to generate anoxic conditions. Sediments were re-suspended for 10 min by mixing an approx. one cm layer using a mechanical mixing device (approx. 150 rotations per minute). Reactors were then left in a dark, cool place for re-sedimentation of suspension over a period of 110 min. After 2 h of exposure, concentrations of P-PO43− were measured in the water. The EPC-0 was determined as the zero of a linear function of Cp-Ck = f(Cp), (where Cp - P-PO43− is concentration prior to exposure and Ck - P-PO43−concentration after exposure) (Wisniewski, 1999; Bartoszek & Koszelnik, 2016). The potential phosphorus load [mgP m−2d−1] releasable from bottom sediments under given conditions was calculated on the basis of the amount of P-PO43− released in ex situ tests, with account taken of the average concentration of this form in the reservoir water throughout the research period.

Results & Discussion

Natural susceptibility to degradation

The parameters of Rzeszów Reservoir affecting natural resilience to degradation are as summarized in Table 1. The Reservoir is assigned to the 4th (or lowest) category of resilience, given its unfavorable hydrological and morphometric parameters, which in fact ensure an almost total lack of resilience to negative impacts from the catchment area. This mainly reflects limited depth and volume of water, and hence a polymictic character of the ecosystem. Equally, the reservoir is characterized negatively by parameters such as the ratio of the active sediment layer surface to the volume of the epilimnion, and Schindler’s ratio. In the case of shallow unstratified reservoirs, bottom sediments across the entire area affect metabolism within the ecosystem as a whole. The catchment area of the reservoir is of 2,060.7 km2, while reservoir volume is just 1.10 Mm3, hence the high value for Schindler’s ratio. The only parameter encouraging a lower risk of eutrophication progressing is the high intensity of water-exchange during the year. This ensures the prevalence of river conditions unfavorable to the development of plankton.

Table 1 Assessment of natural resilience to degradation of the Rzeszów Reservoir, based on Bajkiewicz-Grabowska (1987), Bajkiewicz-Grabowska, (2010).

Parameters	Obtained for Rzeszów Reservoir	Amount	
Average depth (m)	0.6	3	
Ratio of reservoir capacity (‘000 m3) to length of shoreline (m)	0.12	3	
Participation of the water stratification (%)	<20	3	
Ratio of the active sediment layer surface (m2) to volume of epilimnion (m3)	0.46	3	
Intensity of water exchange	456	0	
Schindler’s ratioa (m2 m−3)	1,873	3	
Average value of points	2.5	
Resilience category of the reservoir	IV	
Notes.

a Ratio of the total area of the catchment and the reservoir to the volume of reservoir.

Beyond the morphometric parameters of Rzeszów Reservoir, the geology and nature of the catchment also do much to encourage degradation. Features as summarized in Table 2 speak for assignment to susceptibility category 4 (the worst), denoting a high probability of supply of organic and inorganic matter and nutrients. The probability that critical loads of pollution will be reached is enhanced by the large catchment area in relation to the surface area of the body of water (Ohle’s coefficient), but also by the nature of the flow through the reservoir, the presence of pastureland and agricultural management, and the presence of mountainous terrain conducive to erosion of either natural and anthropogenic origin.

Table 2 Assessment of the Rzeszów Reservoir catchment as a supplier of biogenic and organic matter, based on Bajkiewicz-Grabowska (1987), Bajkiewicz-Grabowska (2010).

Parameters	Obtained for Rzeszów Reservoir	Amount	
Ohle’s coefficienta	4,041	3	
Balance type of lake	flow-through	3	
Average slope in the catchment (‰)	13.8	2	
Geological structure of the catchment	sandy-clayey	1	
Usage of the catchment	pasture - agricultural with buildings	3	
Density of river network (km km−2)	0.41	0	
Contribution of endorheic areas (%)	<20	3	
Average value of points	2.14	
Susceptibility group of the catchment	4	
Notes.

a Ratio of the total catchment area to the reservoir area.

Given assignment to category 4 where both resilience and susceptibility are concerned, it is clear that the setup as regards type of reservoir and type of catchment is maximally negative, and inevitably associated with a considerable risk of eutrophication progressing, due to the adverse natural conditions.

Trophic state and the role of sediments as an internal source of phosphorus

Advanced eutrophication of the waters under study is indicated by values calculated for the Carlson trophic index (Table 3). The calculated TSITP is indicative of hypertrophy, while the TSIChla suggests eutrophic status. Thanks to flow through the reservoir, a part of the phosphorus load is discharged and does not therefore participate in internal processes. This accounts for the above discrepancy between trophic statuses determined by reference to trophic indices for the substrate as opposed to the product. Rzeszów Reservoir resembles other bodies of water recording high values for trophic status (Gruca-Rokosz, Bartoszek & Koszelnik, 2017) in experiencing summer-season oxygen saturation associated with insolation-influenced photosynthesis. However, the related phenomenon of water undergoing alkalization (Bartoszek et al., 2018) has not been detected (Figs. 2A–2B). Concentrations of the substrates nitrogen and phosphorus in reservoir water were relatively high, amounting to 1.46 and 0.171 mg dm−3 respectively, on average. Mean values for the N:P mass ratio below 10:1 indicate that nitrogen is usually present at concentrations limiting the reservoir’s internal production of OM (Galvez-Cloutier & Sanchez, 2007; Hou et al., 2013). A reduction of the ratio of nitrogen to phosphorus (N:P) in water, and therefore stimulation of cyanobacterial bloom, can occur due to the release of phosphates accumulated in sediments (Orihel et al., 2015). It was usual for higher Chla concentrations to be observed in the region of the dam where water flows most slowly (Fig. 2C).

Table 3 Carlson trophic index values (TSI TP, TSI Chla) and selected physico-chemical parameters of water in the Rzeszów Reservoir (E-eutrophy, H-hypertrophy).

Site no.	? n = 9	Tw	TN	TP	Chla	TSITP	TSIChla	
		(oC)	(mg dm−3)	(µg dm−3)			
1	Average	19.3	1.48	0.168	27.1	80	66	
Minim.	13.0	0.88	0.095	1.1			
Max	26.1	2.36	0.268	71.6			
Std. Dev.	4.2	0.4	0.06	27.9	5	8	
Trophic state					H	E	
2	Average	19.4	1.36	0.167	26.6	78	66	
Minim.	13.1	0.82	0.105	3.7			
Max	26.1	1.75	0.282	93.5			
Std. Dev.	4.3	0.3	0.06	28	4	7	
Trophic status					H	E	
3	Average	19.3	1.54	0.177	37.0	80	69	
Minim.	13.2	0.77	0.096	2.5			
Max	26.1	2.43	0.343	161.7			
Std. Dev.	4.6	0.5	0.07	49.5	5	9	
Trophic state					H	E	
Reservoir	Average	19.3	1.46	0.171	30.2	80	67	
Std. Dev.	4.2	0.4	0.06	36	5	8	
Trophic state					H	E	

Figure 2 Variability of oxygen content (%) (A), pH (B) and Chla concentration (C) in the water of the Rzeszów Reservoir.

The abundance of phosphorus compounds in a water depends, not only on the external load supplied, but also on the capacity for internal supply by way of the release of phosphates from bottom sediments under anoxic conditions (Nikolai & Dzialowski, 2014; Bartoszek, 2019; Lee et al., 2019). Phosphorus may be released even where conditions in the near-bottom water are aerobic, on account of the decomposition of OM deposited previously (Dondajewska, 2008; Sobczynski, 2009; Paytan et al., 2017). In the sediments studied, the EPC-0 assumed low values under aerobic conditions, but were approximately 10 times higher in circumstances of anoxia (Table 4). At 0.005–0.108 mgP dm−3, concentrations of P-PO43− in the waters studied were not high at any time during the study period. Only twice was the value for the EPC-0 in aerobic conditions exceeded –at site 1 (near the inlet), with 0.073 and 0.108 mgP dm−3 reported (Figs. 3A–3C). On other dates, release of phosphate from sediment into the near-bottom water was able to occur at the sites studied, with resources depleted by phytoplankton and macrophytes augmented in this way. Hydrobiological analysis carried out in July 2013 showed that the largest number of phytoplankton and at the same time the most cyanobacteria were found at site 3 (near the dam). Cyanobacteria of the genera Microcystis, Oscillatoria, Aphanothece and Romeria have been observed in Rzeszów Reservoir (Bartoszek et al., 2018).

Table 4 The determined EPC-0 values (mgP dm−3) for the bottom sediments of the Rzeszów Reservoirs (r—correlation coefficient).

Site no.	Aerobic conditions	Anoxic conditions	
	EPC-0
(mgP dm−3)	r	n	EPC-0
(mgP dm−3)	r	n	
1	0.071	0.998	6	0.782	0.972	5	
2	0.073	0.996	6	0.529	0.956	6	
3	0.082	0.999	6	0.776	0.943	6	

Figure 3 Variability of phosphate phosphorus concentrations (mgP dm−3) in the water of the Rzeszów Reservoir in relation to EPC-0 values determined for sediments in aerobic conditions.

(A) Site 1; (B) 2; (C) 3.

The stability of phosphorus immobilization in sediments is impacted upon by the types of chemical compound in which the element is present (Bartoszek & Tomaszek, 2011; Dong, Yang & Liu, 2011; Loh et al., 2013). Phosphorus fractionation studies demonstrate that it is the NAIP fraction within TP that is largest (42.5% on average). This reflects combination with Fe, Al and Mn (Table 5). The AP fraction, denoting an association with Ca and a status as least mobile, was at 34.3% on average. The smallest share was taken by the organic fraction –OP (average 21.6%). The average content of mobile phosphorus (NAIP + OP) thus reached 64.1%. For comparison, the NAIP + OP fraction in the bottom sediments of France’s Bort-les-Orgues dam reservoir heavily exposed to anthropogenic influences was found to exceed 80%, including on the basis of NAIP accounting for 59% of TP (Ruban et al., 2001). In contrast, in the sediment of Poland’s Solina Reservoir, the contribution made by NAIP was of only 23.8%, while NAIP + OP stood at 57.7% on average (Bartoszek & Tomaszek, 2011). Solina Reservoir is located in mountainous terrain that is mainly forested or comprising pastureland, while the area is only sparsely populated and developed to a limited extent, with major anthropogenic pressure therefore lacking. The studied samples from Rzeszów Reservoir in fact have a slightly lower content of phosphorus and OM than those from Solina, though they are characterized by approximately 1.5–2 fold lower concentrations of Fe and Al, as well as concentrations of Mn only one-fourth as high (Bartoszek, Tomaszek & Lechowicz, 2015). In contrast, they contained more Ca (approx. 2.5 times), perhaps reflecting the catchment’s mainly agricultural land use. A low value for EPC-0 in the sediments of Rzeszów Reservoir, especially under aerobic conditions, results from the typical mineral nature of the deposits (OM of less than 8%) and the relatively low content of phosphorus with good oxygenation at the interface between bottom sediments and overlying water. By comparison, EPC-0 values in Solina Reservoir ranged from 0.007 to 0.057 mgP dm−3 in aerobic conditions and from 0.103 to 0.169 mgP dm−3 under anoxia (Bartoszek & Koszelnik, 2016). In turn, in a small, heavily-degraded reservoir (at Nowa Wies, SE Poland), EPC-0 values were 0.33 and 4.67 mgP dm−3 (in aerobic and anoxic conditions respectively) (Bartoszek, 2019). EPC-0 values determined around the world under aerobic conditions, for shallow, polymictic lakes and dam reservoirs, ranged from 0.007 to 0.244 and from 0.013 to 0.3 mgP dm−3 (respectively) (Cyr, McCabe & Nürnberg, 2009; Dong, Yang & Liu, 2011).

Table 5 Content of total phosphorus, organic matter (OM), calcium, iron, manganese, aluminum, water (W0) and phosphorus fractions contribution (%) in TP in the bottom sediments of the Rzeszów Reservoir.

Site no.	n = 9	Ca	Fe	Mn	Al	TP	NAIP	AP	OP	OM	W0	
		(mg g−1 of d.w.)	(%)	
1	Average	25.3	22.2	0.598	27.4	0.829	43.3	33.3	21.7	6.89	52.7	
Minim.	20.2	19.1	0.466	22.5	0.668	37.0	28.1	19.1	5.84	44.5	
Max	29.7	29.6	0.697	33.9	1.320	48.9	38.4	25.2	9.07	60.8	
Std. Dev.	3.2	3.2	0.1	3.7	0.19	4.1	2.8	2.1	1.2	5.2	
2	Average	27.4	24.0	0.652	29.8	0.846	41.3	33.7	23.2	7.97	54.9	
Minim.	19.9	19.1	0.487	23.8	0.664	37.5	28.5	18.8	5.28	45.9	
Max	40.3	27.0	0.851	33.2	1.022	46.6	39.1	28.9	10.4	61.3	
Std. Dev.	6.4	2.5	0.1	3.1	0.13	2.9	3.3	3.6	1.8	5.5	
3	Average	25.9	21.0	0.541	25.6	0.711	43.0	35.8	20.0	6.91	48.6	
Minim.	23.1	18.0	0.436	23.2	0.611	37.2	29.7	16.3	5.35	41.7	
Max	27.6	23.8	0.665	27.8	0.863	50.4	41.7	23.2	8.40	57.3	
Std. Dev.	1.5	1.8	0.1	1.3	0.08	5.0	3.6	2.2	0.9	5.0	
Reservoir	Average	26.2	22.4	0.597	27.6	0.795	42.5	34.3	21.6	7.26	52.1	
Std. Dev.	4.1	2.8	0.1	3.3	0.15	4.0	3.3	2.9	1.4	5.7	

Values for EPC-0 differed slightly among the sites studied. Under aerobic conditions EPC-0 increased along the axis of the reservoir, unlike in anoxic conditions. A similar upward trend was observed for the percentage content of the apatite fraction. The release of phosphate from sediments into the water column is mediated by different mechanisms under aerobic and anoxic conditions. Where there is good oxygenation of water it is mainly aerobic decomposition of OM that occurs, with dissolution of the apatite fraction associated with an excess of CO2. However, in the circumstances of a deficiency of oxygen, phosphates are released through anaerobic decomposition of both OM and sparingly soluble inorganic and inorganic-organic compounds. The lowest value for EPC-0 under anoxic conditions was that obtained for the sediments at site 2, despite the fact that it is the deposits in the transition zone of the reservoir that have highest contents of OM and TP. This site also had the sediments with the highest contents of iron, manganese, aluminum and calcium, i.e., elements directly affecting the binding of phosphorus, as well as the highest water content (W0) (Table 5). The smallest contribution to TP of the NAIP fraction, and the largest contribution of the OP fraction also characterized the sediments at site 2. Higher EPC-0 values under anoxic conditions were accompanied by larger NAIP fractions in sediments, with these being more mobile in a situation of low redox potential. No similar trend for the OP fraction could be observed.

The calculated internal phosphorus load can reach an average of 84 mgP m−2d−1 under aerobic conditions and 878 mgP m−2d−1 under anoxic conditions (Table 6). For comparison, the external phosphorus load estimated on the basis of data on the development of the catchment area, i.e., the share of arable land, built-up areas, forests and wastelands, as well as literature run-off coefficients was 266.8 mgP m −2d−1 (Report, 2016). However, this does not take account of actual loads from anthropogenic sources (e.g., industry and services) introduced into the reservoir via tributaries.

Table 6 Estimated value of the internal load in the Rzeszów Reservoir.

Site no	Internal load (mgPm−2d−1)	
	Aerobic conditions	Anoxic conditions	
1	58	662	
2	86	688	
3	107	1,284	
Average	84	878	

Nevertheless, the external load estimated for Rzeszów Reservoir was higher than the calculated internal load under aerobic conditions, but only about one-third as high as under anoxic conditions. The calculated internal phosphorus loads represent possible release in extreme environmental conditions, with resuspension of the surface sediment layer occurring frequently (ex situ for 10 min during a 2-hour test) and oxygen lacking around the clock (under anoxic conditions). In addition, steady phosphate uptake by phytoplankton would have to occur to reduce concentrations in water to the initial level (the average concentration of P-PO43− in reservoir water included in the calculations).

Complete oxygen depletion may occur periodically during a hot and windless summer, but rather in reservoirs with significant sediment enrichment in organic matter and/or at slow water flow (Bartoszek, 2019). The internal load values obtained for lake sediments are much lower (0.16–2.91 and 0.58–27.3 mgP m−2d−1, aerobic and anoxic respectively). However, the values for internal phosphorus load under anoxic conditions are 4–13 times greater than under aerobic conditions (North et al., 2015; Matisoff et al., 2016). Research to date has shown that 1 to 50 mgP m −2d−1 of P can be released from sediment in eutrophic and hypertrophic lakes and reservoirs (Carter & Dzialowski, 2012; Nikolai & Dzialowski, 2014). Due to the different methods and conditions of measurement, it is impossible to compare the results obtained with those of other researchers, who for example did not consider re-suspension of sediment during core-incubation. Researching the sediments of 17 reservoirs in the USA, Carter & Dzialowski (2012) noted that P-release from deposits was usually more intensive in waters that had a higher percentage of arable land in their catchments. Both the budget of P in overlying water and the quantity of release are determined by the inflow of nutrients from external sources (Nikolai & Dzialowski, 2014), which can limit real internal supply significantly in high-flow reservoirs.

Due to intensive mixing and good oxygenation of water (6.9–10.8 mg dm−3), Rzeszów Reservoir has less-favorable conditions for both phosphorus release from sediments and abundant phytoplankton development. However, considerable shallowing and silting increases the possibility of deposit resuspension, which involves an increased area of exchange of substances between sediment and water. One of the undesirable effects of climate change in the temperate zone is the occurrence of ever-longer periods of high temperature in summer, and a decrease in the amount of rainfall, which can lead to periodically significant reductions in reservoir levels, with the result that flows slow and oxygen concentrations decrease. Doig et al. (2017) observed that, at an oxygen concentration of 2 mg dm−3, as opposed to conditions of high oxygenation, there is a more marked internal supply of phosphorus from sediments, albeit one that is still weaker than in anoxic conditions.

Conclusions & Prospects

Due to strong anthropopressure and a number of exceptionally unfavorable morphometric and environmental features, Rzeszów Reservoir is subject to a highly-advanced process of degradation that i.a. does much to limit its utility. Not only the trophic state, but most of all also the possibilities for stored-substance releases and secondary water pollution all attest to the degree to which the reservoir has degraded, most especially on account of its being located in catchment areas under strong anthropogenic influence. However, mineralization of organic matter deposited in the sediments of the reservoir takes place in an undisturbed manner, a fact that attests to its relatively low content, as well as to good oxygenation conditions prevailing in the water.

Obtained EPC-0 values, especially for aerobic conditions, confirm a limited contribution due to internal supply within the phosphate concentration present in the water, this again reflecting good oxygenation and a relatively low content of total phosphorus in sediments. The fact that EPC-0 values are about 10 times higher under anoxic conditions indicates that, in the case of oxygen deficiency, a more intensive process of internal supply of phosphorus can take place. However, since reservoir sediments are poor in organic matter, no major threat of further-progressing eutrophication is likely to be posed, especially in circumstances of strong flow.

It is the bottom sediments in the middle area of the Reservoir that appear most exposed to pollution, as highest contents for most of the parameters determined make clear. The impact exerted by the overall area of the catchment through tributaries and directly overlaps in this part of the reservoir. Closer to the reservoir, the right bank of the direct basin includes housing estates, as well as (in more recent years) an associated network of local roads. These generate pollution via surface runoff. The two attempts at modernizing the reservoir, involving only partial desludging, mainly affected the part near the dam, hence the lower concentration of pollutants in the sediments there.

If the condition of the reservoir is to be improved, desludging and dredging of the object will need to be carried out once again, but this time over a larger area. The removal of an appropriately thick layer of bottom sediment would lead to partial elimination of stored loads of phosphorus and other anthropogenic pollutants (Gulati & Donk, 2002; Wojtkowska, 2013). Determination of that thickness of the layer of sediment needing to be removed would require testing for the accumulation of nutrients and anthropogenic substances in the vertical profile of the deposit, i.e., also in the deeper (5–10, 10–15, 15–20 cm etc.) layers; as well as an increased number of research sites. Liquidation of the shallows would also reduce overgrowth of the reservoir surface by emergent vegetation. With silting of the reservoir occurring so rapidly, action in the overall catchment would also be necessary, to reduce erosion, and hence the inflow of material containing particles of soil, rock fragments and stones.

Supplemental Information

Supplemental Information 1 Assessment system of the susceptibility to degradation

Click here for additional data file.

Supplemental Information 2 Water parameters

Click here for additional data file.

Supplemental Information 3 Phosphorus and its fractions in sediments

Click here for additional data file.

Supplemental Information 4 Other sediment parameters

Click here for additional data file.

Supplemental Information 5 Determination of EPC-0

Click here for additional data file.

We would like to thank our colleagues from the department laboratory for their support and help in sampling and laboratory analysis.

Additional Information and Declarations

Competing Interests

Author Contributions

Data Availability

The authors declare there are no competing interests.

Lilianna Bartoszek conceived and designed the experiments, performed the experiments, analyzed the data, prepared figures and/or tables, authored or reviewed drafts of the paper, and approved the final draft.

Małgorzata Miąsik performed the experiments, analyzed the data, prepared figures and/or tables, authored or reviewed drafts of the paper, and approved the final draft.

Piotr Koszelnik conceived and designed the experiments, analyzed the data, prepared figures and/or tables, authored or reviewed drafts of the paper, and approved the final draft.

The following information was supplied regarding data availability:

The raw measurements are available in the Supplementary Files.

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
