# Peer review of "Trophic degradation predispositions and intensity in a high-flow, silted reservoir"

_PeerJ, doi:10.7717/peerj.9374_

## Round 0.1 · original submission · Major Revisions

Dear Dr. Bartoszek and co-authors,

I just received all the reviews of your manuscript. Although both reviewers consider the study very interesting and providing new findings on the topic, some issues need to be considered before the acceptance.

While reviewer#1 highlights the need to put the importance of your results in a worldwide context, providing new information on phosphorus processes in the reservoir, the reviewer#2 remarks that new relevant literature need to be added in order to improve the quatlity of the results. Numerous errors and inconsistencies must also be corrected. Please, consider all comments and suggestions provide by both reviewers during the revision of your manuscript.

A comprehensive revision of the English of the manuscript is necessary before submitting the new version.

Don't forget to include a letter response along with the revised version of the manuscript. In this letter you must respond point by point to each question.

Best regards,

Salva

Reviewer 1 ·

Basic reporting

The relevance of the study is properly managed, the information is clear and very well presented. It enquires in the deterioration of water quality of small reservoirs associated with high sedimentation of external materials, anthropogenic eutrophication and its relationship with the characteristics of the reservoir. However, I suggest to extent its global importance, for example, adding the number of this kind of reservoirs worldwid. Also, take especial consideration to: 1) Complement the results with external load and output information of P (if available). 2) Discus more about a plausible scenario that favors P realize from sediments at this polymictic shallow system. 3) Please add the Chlorophyll data to figure 2 ( if available) .

Experimental design

The description of the study area is complete. It covers the relevant aspects of the reservoir. I would recommend, if data are available, to add external P loads and outputs, sedimentation rates over time, especially after dredging operations.

Validity of the findings

3 Results
Results of natural susceptibility to degradation do to its morphometric and hydrological characteristics and its trophic state are clearly stated. However, at the introduction mentioned the great relevance of sedimentation of suspended inflowing material and the allochthonous river-sediment transport to water quality and in the results this aspect seems lost. I highly recommend to add more information about inputs and outputs of P during time and then compare them with the possible internal source.
A relevant result is the difference between EPC-0 under aerobic and anoxic conditions. However, your oxygen results do not suggest an anoxia scenario could take place at this reservoir. I was wondering if with these results is possible to estimate a more realistic oxygen concentration that enhance the release of phosphorous. Could you give an estimation of the potential internal load Phosphorous?
4 Tables
In general, they are appropriate, I only suggest to make a more notorious division between columns and units, because they could be a little bit confusing, especially in table 5.
I recommend that table 5 should be edited by changing de TP column before the percentage of species of P in order to make it more comfortable for readers.

5 Figures
The figures looks fine, I would recommend to ad chlorophyll graph to figure 2 to complement and strengthen the results.

5 Conclusions & Perspectives
First paragraph visibly explains how the morphological and hydrological characteristics of the reservoir and anthropogenic activities prone deterioration of water quality. Even second paragraph is well structured, it seems not concluding results at all. Does the obtained data can be used to estimate the amount of sediments that should be dragged in order to improve the water quality?

Additional comments

In general, the paper is enjoyable to read, it is well structured and written. However, I recommend to double check it specially, units (e.g. Line 87 and line 167) and to change the writing of line 71 to 75, because it is quite similar to the abstract.

Reviewer 2 ·

Basic reporting

Review of Manuscript Number: #44688
The manuscript "Predispositions and intensity of trophic degradation of the highly flow, silted reservoir" by L. Bartoszek et al. presents the results of research on trophic degradation of Rzeszów Reservoir as an example of small reservoir located in areas with high anthropopressure. The large number of this type of water reservoirs and numerous problems related to maintaining good quality of water collected in them motivate us to conduct continuous research in this area. The authors' comprehensive view on factors influencing the trophic degradation of the studied reservoir should be emphasized. The research were carried out based on standard methods, but results are interesting. Unfortunately, these manuscripts require significant corrections and supplementation by literature in particular "Introduction" and presentation of results obtained against the background of broader literature that goes beyond Polish literature.
Reviewer recommendation: Major revision

Experimental design

The natural susceptibility to degradation water reservoir was evaluated on the base of two procedures (line 112-117) by Bajkiewicz-Grabowska 1987, 2010;and Markowski, Kwidzinska 2015. This works are published in English,but are known and used generally only in Poland. Both procedures should be thoroughly described.

Validity of the findings

no comment

Additional comments

Review of Manuscript Number: #44688
The manuscript "Predispositions and intensity of trophic degradation of the highly flow, silted reservoir" by L. Bartoszek et al. presents the results of research on trophic degradation of Rzeszów Reservoir as an example of small reservoir located in areas with high anthropopressure. The large number of this type of water reservoirs and numerous problems related to maintaining good quality of water collected in them motivate us to conduct continuous research in this area. The authors' comprehensive view on factors influencing the trophic degradation of the studied reservoir should be emphasized. The research were carried out based on standard methods, but results are interesting. Unfortunately, these manuscripts require significant corrections and supplementation by literature in particular "Introduction" and presentation of results obtained against the background of broader literature that goes beyond Polish literature.
Reviewer recommendation: Major revision
Specific comments
The literature used in this article should be supplemented.
Line 37-38: "...,as associated with a rapid reduction in capacity" - too much simplification, please expand.
Line 50-52: no reference to literature.
Line 61-63: EPC-0 - the most common abbreviation of Equilibrium Phosphorus Concentration is EPC0. The EPC0 is defined as the concentration of soluble reactive phosphorus (SRP) in solution which, when placed in contact with sediment, produces no net sorption or desorption of soluble reactive phosphorus (SRP) over a period of 24 h. The EPC0 method was presented much earlier than the cited literature. Please provide relevant literature.
Line 87: Please indicate the initial and current Rzeszow Reservoir volume and its percentage change.
Line 97-101: Please specify the slope of the river bed from its source to the reservoir. Difference in water level in meters don't give information. The structure of land use in the reservoir's catchment area should be supplemented too.
Line 112-117: The natural susceptibility to degradation water reservoir was evaluated on the base of two procedures known only in Poland. Although these (Bajkiewicz-Grabowska 1987, 2010; Markowski, Kwidzinska 2015) works are published in English, are not cited in the wider literature. Both procedures should be discussed in more detail and thoroughly described.
Line 131: SMT ?? All abbreviations should be explained before their first use.
Line 190-192; 193-195: no reference to literature.

---

## Round 0.2 · accepted · Accept

Dear Dr. Bartoszek and co-authors,

Your revised manuscript improved a lot and now it is ready to be accepted.

Congratulations!

Reviewer 1 ·

Basic reporting

Nothing to add

Experimental design

Nothing to add

Validity of the findings

Nothing to add

Additional comments

Thanks to taking into account the comments of reviewers, the manuscript reflects more clearly your hypothesis. Congratulations for your work.

Reviewer 2 ·

Basic reporting

Review of Manuscript Number: #44688
The revised version of the manuscript "Trophic degradation predispositions and intensity in a highflow, silted reservoir" (earlier "Predispositions and intensity of trophic degradation of the highly flow, silted reservoir") by L. Bartoszek et al. is improved. The overall coherence of the paper and clarity has visible increased. The simplifications used in the manuscript have been developed. References was supplemented. Methods used by Authors was detailed clarified in manuscript and supplements. On this basis, I recommend the manuscript for publication.

Experimental design

no comment

Validity of the findings

no comment

Additional comments

no comment